# Social–Emotional Competence and Academic Achievement of Nursing Students: A Canonical Correlation Analysis

**DOI:** 10.3390/ijerph18041752

**Published:** 2021-02-11

**Authors:** Sun-Hee Kim, Sujin Shin

**Affiliations:** 1College of Nursing, Research Institute of Nursing Science, Daegu Catholic University, Daegu 42472, Korea; sunhee421@cu.ac.kr; 2College of Nursing, Ewha Womans University, Seoul 03760, Korea

**Keywords:** professional competence, academic success, emotional intelligence, students, nursing

## Abstract

This study was conducted to investigate the correlations between social–emotional competence (SEC) and academic achievement (AA) among nursing students and to compare students’ level of each core skill of SEC (critical thinking disposition, self-directed learning, creativity, emotional intelligence, problem-solving, and collaboration) and academic achievement (clinical performance and subjective academic achievement). A cross-sectional design was adapted. Data were collected from 195 nursing students in the junior and senior years who had participated in clinical practicum from four universities in South Korea. General characteristics, levels of critical thinking disposition, self-directed learning, creativity, emotional intelligence, problem-solving, collaboration, and academic achievement were collected via self-reported questionnaire. Canonical correlation analysis was performed to evaluate the relationship between SEC and AA. The canonical correlation coefficient between SEC and AA was 0.762. Critical thinking disposition (Rs = 0.89), problem-solving (Rs = 0.86), and cooperation (Rs = 0.80) made the most important contributions to SEC. Clinical performance (Rs = 0.95) and subjective AA (Rs = 0.57) were correlated with AA. SEC should be addressed to improve the AA of nursing students. All core skills of SEC should be regularly promoted. It is particularly urgent for nursing students to improve their creativity.

## 1. Introduction

The highly anticipated fourth industrial revolution is expected to bring massive data and artificial intelligence-driven smart information technology to every corner of our lives [1]. To secure competence in this intelligent information-based society, prospective workers should be able to acquire new knowledge and efficiently make use of it, and should also be equipped with the unique forms of social–emotional competence (SEC) that cannot be replaced by robots [1]. Furthermore, the increasing diversity in terms of students’ race and socioeconomic background in today’s educational settings requires a greater focus on nurturing abilities such as relationship-building, emotional management, and execution, which allow smooth communication and collaboration with peers from diverse backgrounds [2]

The 2016 World Economic Forum suggested that critical thinking, problem-solving, creativity, communication, and collaboration would be the core skills of the 21st century for dealing with complicated challenges. Curiosity, initiative, persistence/grit, adaptability, leadership, and social and cultural awareness were presented as the core character qualities that define how students approach their changing environment. SEC features many different skills and character components required in the 21st century that strengthen academic performance and professional readiness [3] and reduce the possibility of problematic behavior and emotional pain [2,3], making SEC crucial to jobs in the future [1].

Given these demands of the new era, the South Korean educational system has focused on motivating students to take initiative in their studies over the past decade [4]. However, many teenagers still seem to have low SEC, lacking healthy self-awareness and showing poor social interactions and self-understanding, low self-respect and self-control, and low levels of social qualities such as volunteering, empathy, communication, and open-mindedness [5]. This is a result of providing insufficient education for developing SEC in school curricula that are preoccupied by a hierarchical college ranking system and the race for getting better grades to enter the best schools. Research shows that social–emotional learning competence improved when nursing students participated in social activities [6], suggesting that SEC can be fostered through education and training [7] and that it has become a core skill set for nurses. This implies that nursing education should focus on cultivating talented individuals who are suitable for the smart information society that has become a reality and training them to be more competent in terms of creativity, problem-solving, self-management and collaboration, and emotional competence.

Nursing students hone their nursing skills through education and training as they prepare to enter their profession. Clinical performance is one of the key elements used to evaluate the quality of nursing education. The academic accomplishments of undergraduate students are affected not only by individuals’ cognitive intelligence, but also by SEC [3]. Clinical performance and subjective academic achievement (AA) should be included as evaluation criteria for academic performance to evaluate students’ level of preparation and academic readiness to become a nurse.

Among the 16 important SECs of 21st century education announced at the world economic forum, critical thinking, problem-solving, creativity, collaboration, and adaptability (the ability to process emotion) [1], and among the dimensions of the Collaborative to Advance Social and Emotional Learning (CASEL) model, self-awareness, self-management, and relationship skills [2], were integrated to construct the SEC components of this study. Critical thinking, problem-solving, creativity, and a part of emotional intelligence were constructed as a dimension of self-awareness, which represents the accurate recognition of one’s own thoughts and emotions and the influence of one’s thoughts and emotions on behavior [2]. A part of emotional intelligence and self-directed learning were constructed as a self-management dimension that means the effective control of one’s thoughts, emotions, and behaviors [2]. Collaboration was constructed as a dimension of relationship skills, which explains building and maintaining relationships, seeking help and providing help when necessary, and communicating effectively. Therefore, the components of SEC were critical thinking, problem-solving, creativity, emotional intelligence, collaboration, and self-directed learning in this study. Academic achievement has been reported to be related to critical thinking [8,9], problem-solving [10], creativity [11], emotional intelligence [12,13,14,15], collaboration [16,17], and self-directed learning [8,10,17].

Existing research on SEC among nursing students has explored partial correlations between some elements of SEC skills such as critical thinking [8,9], problem-solving [10], creativity [18], emotional intelligence [12,13,14,15], collaboration [19], and self-directed learning [8,10,18]. However, limited research has sought to analyze correlations between SEC and AA among nursing students in a comprehensive manner, as well as the relative importance of core SEC skills and correlations between the application of those core skills and academic performance. As we evolve into a smart information-based society, it is imperative to confirm such correlations in order to propose new directions for nursing education that can meet changing demands.

The purpose of this study was to investigate the correlations between SEC and AA among nursing students. The more detailed goals were as follows: first, to gain an understanding of levels of SEC (critical thinking, problem-solving, creativity, emotional intelligence, collaboration, and self-directed learning) and AA (clinical performance and subjective AA) among nursing students; second, to find correlations among critical thinking, problem-solving, creativity, emotional intelligence, collaboration, self-directed learning, clinical performance, and subjective AA among nursing students; and third, to establish canonical correlations between SEC variables and AA variables among nursing students.

## 2. Materials and Methods

### 2.1. Design

This research was a cross-sectional descriptive survey conducted to find correlations between SEC and AA among nursing students.

### 2.2. Subjects

The subjects of this study were junior and senior undergraduate nursing students who had experienced clinical training and voluntarily agreed to participate in the study. The sample size was calculated using G*Power 3.1.9.2 (Heinrich-Heine-Universität, Düsseldorf, German). The minimum sample size needed to analyze the relevant correlations was 191 when the effect size was medium (0.2), significance level was 0.05, and statistical power was 0.80. The recruitment goal was 200 subjects, considering the possibility of drop-out and withdrawal from participation, and the study ultimately included 195 subjects.

The general characteristics of the participants were shown Table 1. In total, 90.3% of the participants were female, 51.8% were juniors and 48.2% were seniors, and 51.8% of the participants had received clinical training for two or fewer semesters and 48.2% had received clinical training for three or more semesters.

### 2.3. Measure

#### 2.3.1. Social–Emotional Competence

Social–emotional competence (SEC) was measured including critical thinking disposition, self-directed learning, creativity, emotional intelligence, problem-solving, and collaboration abilities.

Critical thinking disposition. Critical thinking disposition was measured using Yoon’s Critical Thinking Disposition (YCTD) instrument for nursing students [20]. The seven subscales of the YCTD include healthy skepticism (four items), intellectual fairness (four items), objectivity (three items), systematicity (three items), prudence (four items), intellectual eagerness/curiosity (five items), and critical thinking self-confidence (four items). Two (4, 14) of the 27 items are scored using reverse coding, with a higher score on a five-point Likert scale indicating a greater disposition towards critical thinking. The instrument had a Cronbach’s α of 0.905 in this study. The internal reliability of seven subscales ranged 0.610–0.796.Self-directed learning. Self-directed learning was measured using a tool for college students and adults developed by Lee et al. [21]. The tool consists of three ability factors and eight sub-categories, with 45 detailed items including 20 items to measure study planning ability (diagnosing the desire to study, study goal setting, and understanding study resources), 15 items to measure actual fulfillment of studying (basic self-management skills, selecting study strategies, and persistence), and 10 items to measure study evaluation ability (efforts/outcomes and self-reflection). Each item was measured on a Likert scale from 1 to 5, with 1 being “very rarely” and 5 being “very frequently”, and negative items with reverse points. The higher the score, the higher the self-directed learning. The reliability measured by Cronbach’s α was 0.904, and Cronbach’s α for subscale ranged 0.568–0.811 in this study.Creativity. Creativity was measured using an adapted version of Runco’s Ideational Behavior Scale (RIBS) [22] for college students with enhanced credibility and validity, as developed by Lee and Lee [23]. This tool consists of a total of 23 items regarding how an individual utilizes and deals with ideas. Each item is measured on a five-point Likert scale with 1 being “not at all” and 5 being “extremely”. Higher scores indicate a better ability to utilize and create ideas. The Cronbach’s α was 0.932 in this study.Emotional intelligence. Emotional intelligence was measured using the Wong and Law Emotional Intelligence Scale (WLEIS) adapted by Jung and Jung [24]. The tool consists of 16 items in four ability dimensions of appraisal and expression of emotion in the self (four items), appraisal and recognition of emotion in others (four items), regulation of emotion in the self (four items), and use of emotion to facilitate performance (four items). Each item is measured on a seven-point Likert scale, with 1 being “not at all” and 7 being “extremely”. The higher the score, the higher the emotional intelligence. The Cronbach’s α obtained a value of 0.936 and Cronbach’s α for subscale ranged 0.691–0.800.Problem-solving. Problem-solving was measured using a tool developed by Lee et al. [25] to measure the problem-solving process of adults. The tool consists of five subcategories that include clarifying the problem (six items), seeking a solution (six items), making a decision (six items), applying the solution (six items), and evaluation and reflection (six items). The responses for each of the 30 items are recorded on a five-point Likert scale, with 1 being “not at all” and 5 being “extremely”. Higher scores indicate a higher ability for problem-solving. In the present study, the tool had an overall Cronbach’s α of 0.957, and subscales for 0.830–0.881.Collaboration. Collaboration was measured using a tool developed by Chung [26]. The tool consists of 17 items, divided into eight items for relationship teamwork and nine items for technical teamwork. The responses for each of the 17 items are recorded using a five-point Likert scale, with 1 being “not at all” and 5 being “extremely”. Higher scores indicate a higher ability for collaboration. In the present study, the tool was found to have a Cronbach’s α of 0.845 for human relationship teamwork and 0.859 for technical teamwork, and an overall Cronbach’s α of 0.941.

#### 2.3.2. Academic Achievement

Academic achievement (AA) was measured from clinical performance and subjective academic achievement.
Clinical performance. Clinical performance was measured with a tool developed by Lee et al. [27] and revised by Yang and Park [28]. The tool consists of 19 items in six subscales including nursing procedures (four items), nursing intermediation (four items), socio-psychological nursing (three items), education (three items), physical check-up and patient monitoring (two items), and basic nursing (three items). The responses for each item are recorded on a five-point Likert scale, with 1 being “not at all” and 5 being “extremely”. Higher scores indicate a higher ability for clinical performance. The Cronbach’s α of this tool was 0.925 in the present study.Subjective academic achievement (AA). Subjective AA indicates the level of self-perceived AA. In this study, a visual analogue scale of 0 to 100 was used, allowing students to mark their own level of achievement. The higher the points were, the higher they perceived their AA.

### 2.4. Data Collection

The study was approved by the Daegu Catholic University Institutional Review Board (CUIRB-2018-0009) prior to data collection. Data collection was carried out after notifying participants that the data would be collected through an online survey. The researcher selected four nursing departments located in three urban cities and asked the deans for cooperation for data collection, providing information on the purpose and methods of research. One nursing student was selected as a research assistant for each university, and a research assistant was given instructions about the purpose of the research, data collection method, confidentiality and anonymity, the withdrawal policy, and possible benefits and disadvantages. The survey was conducted among nursing undergraduates who acknowledged that they understood the instructions and voluntarily agreed to participate in the research. Participants who completed the survey were rewarded with small gift.

### 2.5. Data Analysis

The frequency, percentage, average, standard deviation, internal consistency, and reliability (Cronbach’s α) of the participants’ general background information were analyzed using SPSS Statistics 25.0 (IBM Corp., Armonk, NY, USA). Pearson correlation analysis was used to analyze the correlations between the SEC variables (critical thinking disposition, problem-solving, creativity, emotional intelligence, collaboration, and self-directed learning) and the AA variables (clinical performance and subjective AA). Canonical correlations between the SEC variables and the AA variables were analyzed after standardizing the scores of each variable.

### 2.6. Ethical Considerations

The Helsinki Statement was fully observed for ethical considerations and to ensure that the research was ethical for the subjects. The survey was conducted after receiving approval from the college Institutional Review Board. Furthermore, all participants were provided instructions, including information on the purpose and methods of research, withdrawal from the survey, and ethical protections such as confidentiality and anonymity. The survey was administered among participants who agreed to the terms.

## 3. Results

### 3.1. Social–Emotional Competences and Academic Achievement Levels

Table 2 illustrates the result of comparisons between SEC variables and AA variables, based on a 100-point scale for each area. For SEC, a score of 80.8 was reported for collaboration, 72.8 for problem-solving, 72.7 for emotional intelligence, 71.0 for critical thinking disposition, 67.6 for self-directed learning, and 59.4 for creativity. The average scores for the AA variables were 72.6 for clinical performance and 63.5 for subjective AA.

### 3.2. Correlations between Social–Emotional Competences and Academic Achievement

The analyses of SEC and AA showed that the correlations ranged between r = 0.295 and r = 0.646 for all sub-variables other than creativity and subjective AA, indicating there was a moderate level of correlation (Table 3).

For the canonical correlation analysis, the X variables represented SEC (X_1_ = critical thinking disposition, X_2_ = self-directed learning, X_3_ = creativity, X_4_ = emotional intelligence, X_5_ = problem-solving, and X_6_ = collaboration) and the Y variables represented AA (Y_1_ = clinical performance and Y_2_ = subjective academic achievement). The two formulas presented below were derived through the canonical correlation analysis.

These are equations:X = 0.263χ_1_ + 0.216χ_2_ − 0.016χ_3_ + 0.166χ_4_ + 0.236 χ_5_ + 0.349 χ_6_(1)
Y = 0.853y_1_ + 0.339y_2_(2)

The X variables and Y variables were confirmed to represent SEC and AA, with percentages of 58.5% and 61.1%, respectively. The Wilks coefficient for the canonical correlation between the two variable groups proved meaningful at 0.394 (F = 15.332, *p* < 0.001) and the canonical correlation coefficient was 0.762 (Figure 1), which is large effect size. The variables with the highest correlation with SEC were critical thinking disposition (Rs = 0.89), problem-solving (Rs = 0.86), and collaboration (Rs = 0.80), while AA had the highest correlation with clinical performance (Rs = 0.95).

## 4. Discussion

This study was conducted to explore the correlation between SEC and AA among nursing undergraduates in the fourth industrial revolution and to understand the relative importance and level of the components of SEC and AA. The canonical correlation between SEC and AA was 0.762 in this study. This means that nursing students with high SEC have better clinical performance and subjective AA, which leads to a generally high AA. This result is in line with that of previous studies, which suggested that higher SEC leads to higher AA [29,30]. Critical thinking disposition and self-directed learning and communication ability affect clinical performance [8], and emotions are correlated with AA [15]. In other words, this study provided evidence for the need to reinforce SEC in nursing students in order to boost their AA by proving a more comprehensive canonical correlation among all elements of SEC and AA compared to existing studies that only investigated some elements of SEC and AA.

However, the fact that the curriculum for nursing students requires relatively little imagination and emotional sensitivity [31] suggests that SEC needs to be developed separately. In the future, nurses should be able to resolve problems requiring creativity and critical thinking and provide emotional support to patients, which are areas that cannot be replaced by artificial intelligence. Nursing departments should therefore develop both an overt curriculum and a hidden curriculum that foster SEC among nursing students.

Among the six elements of SEC measured in this study, critical thinking disposition and problem-solving were most closely correlated with SEC among nursing students, while nursing students were found to have relatively low levels of critical thinking disposition and problem-solving. This indicates that the various educational strategies currently being utilized to nurture these important critical thinking abilities among nursing students are not playing a sufficient role. Given that critical thinking disposition and problem-solving are related to clinical performance [8,9,10] and that problem-solving could be improved through critical thinking education [32], it is necessary to develop a way to improve critical thinking ability and to implement critical thinking education.

Collaboration is needed in any profession, but is particularly crucial for nurses [33]. This study found that collaboration was the most important factor in SEC, with a score of 80.8 out of 100. Collaboration is a complex process that requires sharing, respect, and teamwork [34], and communication ability is an influential factor in collaboration [35]. It is therefore important to improve communication among team members. Students’ experience of collaborative study such as clinical training and group assignments in the curriculum is thought to have positive effects on building collaborative ability. Problem-based learning [36] and standardized patient (SP) education strategies [19] that use students to improve collaboration and collaborative testing improve academic achievement [16], therefore a variety of collaboration strategies are needed for nursing education. In addition, collaboration across professions has become equally important as collaboration within the nursing profession for providing the best results for patients [37], and it will be necessary to introduce cross-professional education into the nursing curriculum and study its effects.

Emotional intelligence was the fourth most important factor of SEC among nursing students. Nurses with high emotional intelligence take care of patients proactively, treat them with a calm and confident attitude, communicate with precise language, and tactfully deal with and manage patients in various circumstances [14]. Emotional intelligence is also related to clinical performance [12,14,38], and nursing students who participated in programs to enhance emotional intelligence were found to have improved their communication ability, interpersonal relationships, resilience, stress management, clinical performance, and internal locus of control [13]. This indicates that emotional intelligence is an important factor that should be considered in designing nursing curricula.

Self-directed learning was the fifth most relevant component of SEC among nursing students in this study, with a relatively low score of 67.6 out of 100. Korean nursing students have low self-directed learning, since their education is focused on rote learning and memorizing knowledge, as well as competition to get better grades [4], even though two decades of educational reform have sought to foster creativity and problem-solving. Many studies have suggested that self-directed learning of nursing students is highly relevant to clinical performance [8,10]. Improving self-directed learning should be supported by structural and systematic changes across Korean society. Nursing professors should also provide more opportunities for self-directed learning.

This study found creativity to be the least important component of SEC, scoring 59.4 out of 100. Creativity is a required skill for nurses who need to take care of many different patients, are often faced with unexpected situations, and cannot satisfy the demands of diverse types of patients by simply performing daily practices. Therefore, creativity is a necessary skill for a nurse. Diversity learning [39,40], freedom to learn [39], self-directed learning [18], learning with confidence [39], and learning through group work [32,39] help improve creativity. Innovative pedagogy should support students’ creativity and creative problem-solving [39,41], in light of findings that the instructor’s quality and a teaching style of accepting students’ requests [41] and a creative school environment are perceived by students lead to enhanced creativity [40,42]. Considering this, instructors should consider ways to help students foster creativity in their pedagogical strategies and study environment.

Clinical performance and subjective achievement explained 61.1% of AA among nursing students, and students with higher SEC were proven to have higher clinical performance and AA. However, in a clinical study environment focused on observing how others practice rather than on getting hands-on experience [43], opportunities to develop one’s own clinical performance can be limited. Feedback is crucial for promoting effective learning in a clinical training environment that lacks first-hand experience. Providing constructive feedback to students is one of the key aspects of education that enhance the learners’ experience, and it is important for both learners and instructors. However, it has been reported that students are not receiving as much feedback as they would like [44,45]. Instructors should construct a curriculum that fits the learning goals of students and productive evaluations should be improved by supporting preceptors.

Subjective AA was proven to be an important factor for overall AA, scoring 63.5 out of 100. Students’ self-confidence was also a determining factor for AA [46], and motivational processes had both an impact on AA itself and intermediary effects between emotions and AA [15]. Using blended problem-based learning strategies helped improve AA among nursing students [47], and encouraging a learning style appropriate to individual learners, such as discussions, lecture, skill training, and project execution also showed a relationship with better AA [48]. In order to improve AA, it is necessary to create a learning environment that enhances self-confidence, positive emotions, and motivational processes and to select class plans and pedagogical strategies in consideration of the key learning styles of each class.

This study is meaningful in that it demonstrated the canonical correlation between SEC and AA of nursing students, with comprehensive analyses that factored in the various elements of SEC. This study confirmed correlations among the six elements of SEC that have been reported to be the core skills of the fourth industrial revolution era, including critical thinking disposition, self-directed learning, creativity, emotional intelligence, problem-solving, and collaboration. The analysis also found a relative order of importance among the elements for nursing students, and that creativity was the most unmet skill relative to its importance. Subsequent steps should include designing a curriculum that prioritizes the development of key skills that are unmet based on the relative weights derived from the results of this study. Since this study measured AA using two criteria—clinical performance and subjective AA—the results should be interpreted with caution. Furthermore, since this study surveyed nursing students in some regions of Korea, it is difficult to generalize the results of this study. Further research is needed to improve our understanding of students’ actual competence, such as objective evaluations of clinical performance and grades.

## 5. Conclusions

The study found correlations between SEC and AA among nursing students. Elements of SEC, including critical thinking disposition, problem-solving, collaboration, self-directed learning, creativity, and emotional intelligence, should be fostered to enhance the academic performance of nursing students, in that order. It is particularly urgent to nurture creativity among nursing students, given that it is one of the most crucial professional skills in the 21st century. Future studies should further explore the main factors of SEC and AA, as well as the immediate impact of nurturing the main factors of SEC on AA.

## Figures and Tables

**Figure 1 ijerph-18-01752-f001:**
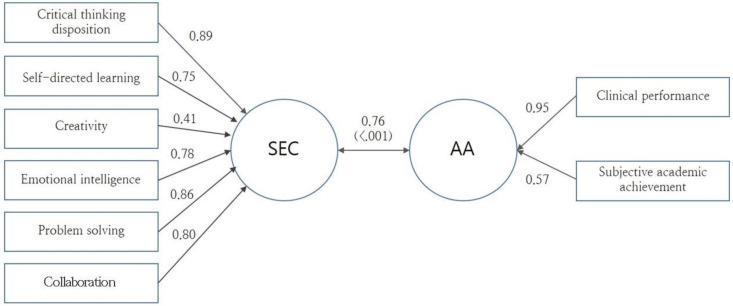
Structure coefficient of canonical factors. SEC: social emotional competencies; AA: academic achievement.

**Table 1 ijerph-18-01752-t001:** General characteristics (*n* = 195).

Variables	Categories	*n*	%	Mean ± SD
Age (years)				21.86 ± 1.59
Gender	Male	19	9.7	
	Female	176	90.3	
School year	Junior	101	51.8	
	Senior	94	48.2	
Number of semester taking clinical practicum course	1–2	101	51.8	
	≥3	94	48.2	

**Table 2 ijerph-18-01752-t002:** Descriptive statistics of social–emotional competencies and academic performance (*n* = 195).

	*n*	Possible Range	Min	Max (A)	Mean (B)	SD	B×100/A
**Social**–**emotional competencies**							
Critical thinking disposition	192	27–135	56.00	129.00	95.87	11.05	71.0
Self-directed learning	189	45–225	70.00	205.00	152.06	18.34	67.6
Creativity	190	23–161	34.00	141.00	95.69	19.52	59.4
Emotional intelligence	190	16–112	48.00	112.00	81.45	13.54	72.7
Problem solving	194	30–150	30.00	150.00	109.30	15.74	72.8
Collaboration	193	17–85	46.00	85.00	68.74	8.86	80.8
**Academic performance**							
Clinical performance	179	19–95	47.00	95.00	68.93	9.13	72.6
Subjective academic achievement	194	0–100	0.00	100.00	63.52	17.28	63.5

**Table 3 ijerph-18-01752-t003:** Correlation between the sub-dimension of social emotional competencies and academic performance (*n* = 195).

Variables	χ_1_	χ_2_	χ_3_	χ_4_	χ_5_	χ_6_	y_1_	y_2_
Critical thinking disposition (χ_1_)	1	0.659 (<0.001)	0.491 (<0.001)	0.661 (<0.001)	0.812 (<0.001)	0.542 (<0.001)	0.646 (<0.001)	0.370 (<0.001)
Self-directed learning (χ_2_)		1	0.384 (<0.001)	0.515 (<0.001)	0.636 (<0.001)	0.373 (<0.001)	0.497 (<0.001)	0.432 (<0.001)
Creativity (χ_3_)			1	0.298 (<0.001)	0.491 (<0.001)	0.150 (0.055)	0.328 (<0.001)	0.106 (0.179)
Emotional intelligence (χ_4_)				1	0.554 (<0.001)	0.593 (<0.001)	0.553 (<0.001)	0.370 (<0.001)
Problem solving (χ_5_)					1	0.544 (<0.001)	0.630 (<0.001)	0.350 (<0.001)
Collaboration (χ_6_)						1	0.595 (<0.001)	0.295 (<0.001)
Clinical performance (y_1_)							1	0.274 (<0.001)
Subjective academic achievement (y_2_)								1

## Data Availability

The data presented in this study are available on request from the corresponding author. The data are not publicly available due to legal and privacy issues.

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
