# Peer review of "Social–Emotional Competence and Academic Achievement of Nursing Students: A Canonical Correlation Analysis"

_ijerph, 2021, doi:10.3390/ijerph18041752_

Round 1

Reviewer 1 Report

Dear Authors, this interesting manuscript provides information to assess the correlations between social-emotional competence (SEC) and academic achievement (AA) among nursing students and to compare students’ level of each core skill of SEC The study design was appropriate to answer the aim. i would suggest just to enhance the discussion and enrich this part with more findings from the international literature. It will be a very interesting manuscript for the readers.

Author Response

Dear Authors, this interesting manuscript provides information to assess the correlations between social-emotional competence (SEC) and academic achievement (AA) among nursing students and to compare students’ level of each core skill of SEC The study design was appropriate to answer the aim. i would suggest just to enhance the discussion and enrich this part with more findings from the international literature. It will be a very interesting manuscript for the readers.

→ We appreciate the time and effort that you have put into the valuable feedback provided on this manuscript. We added the discussion from international literature statements to enhance and enrich the discussion section (page 8-9)

Reviewer 2 Report

Title:
Too long. It is recommended to reduce the number of words

Participants

The characteristics of the participants are best included in the "subjects" section rather than in the results section.
What was the purpose of finding out participants' religious beliefs?

Instruments:

It would be interesting to include Cronbach's alpha for each of the sub-scales of those instruments that have them.

What work did the hired research assistant do, and what was his or her background?

There is little discussion of the limitations of the study and these need to be developed further.

A thorough review of the references is needed; there are several flaws.

Author Response

Title: Too long. It is recommended to reduce the number of words

→ We revised it to reduce the length of the title as follows: Social-Emotional Competence and Academic Achievement of Nursing Students: A Canonical Correlation Analysis

Participants: The characteristics of the participants are best included in the "subjects" section rather than in the results section. What was the purpose of finding out participants' religious beliefs?

→ We moved Table 1 from the results section to the subject section, and the contents of religious beliefs were deleted because they were not related to the purpose.

Instruments: It would be interesting to include Cronbach's alpha for each of the sub-scales of those instruments that have them.

→ We presented Cronbach's alpha for each of the sub-scales in Critical thinking Disposition, Self-directed Learning, Emotional Intelligence, and Problem-solving.

What work did the hired research assistant do, and what was his or her background?

→ The sentence was corrected as it was misleading.

There is little discussion of the limitations of the study and these need to be developed further.

→ We mentioned the limitations of this study on page 10.

A thorough review of the references is needed; there are several flaws.

→ We reviewed the reference list by referring to the Instruction for Author and revised several flaws.

Reviewer 3 Report

This study analyzes the correlations between socio-emotional competence (SEC) and academic performance (AA) among nursing students and compares the students' level of each SEC core skill (critical thinking disposition, self-directed learning, creativity, emotional intelligence, problem solving and collaboration) and academic performance (clinical performance and subjective academic performance), with the ultimate goal of reflecting on a necessary educational program at SEC for nursing students. It is a work as interesting as it is necessary, however, it could be improved. Specifically, I propuse the following recommendations for publication. Regarding the introduction, given that the central axis of the work is to justify the need to acquire socio-emotional competence (SEC) for nursing students, I consider it necessary to explain how the authors conceptualize the socio-emotional competence construct (SEC) on which they plan the study. We verify throughout the work that they incorporate different measures as components of the same, but they do not clarify why these and not others, nor do they indicate any theoretical model that encompasses them as part of the same construct. On the other hand, scientific evidence indicated in this section on which the study is justified are not referenced, for example, the paragraph that begins on line 68. They are developed in the discussion. Regarding the description of the evaluation instruments, and in coherence with what I have just indicated, it is not clear why different instruments are used to measure a single construct. On what scientific and / or theoretical basis do you consider that the different variables evaluated correspond to the SEC construct. It would be necessary to explain the model on which they are based to indicate that the SEC is made up of the dimensions that they indicate as defining it. On the other hand, in all the instruments described, it does not define what it actually measures, not how it obtains the measure it uses. In almost all of its describes that they are made up of different subscales, of which sometimes it reports Cronbach's α and other times it does not, or vice versa (as it does with the collaboration variable). You would have to indicate how you get the only punctuation you use, its meaning and its alpha. Regarding the results section, given the imbalance in the number of components of some categories, and given the possible influence of these variables on the variables studied, it would be necessary to include them in the correlation analyzes and perform analysis of variance Regarding the results section, given the imbalance in the number of components of some categories, and given the possible influence of these variables on the variables studied, it would be necessary to include them in the correlation analyzes and perform analysis of variance to check if there are statistically differences significant between the qualitative variables (for example gender and religion) and the dependent variables, reporting in both cases the effect size and specifying which in "small", "medium" and "large". statistically differences significant between the qualitative variables (for example gender and religion) and the dependent variables, reporting in both cases the effect size and specifying which in "small", "medium" and "large". Likewise, regarding the Discussion section, in coherence with what was commented in the introduction, it is striking that it is where more articles are incorporated that link the variables studied in the population studied. These articles should be transferred to the introduction to justify the reason for the proposed research, and the reason for the incorporation in the study of the variables studied in the type of students studied (nursing). That is, we find more studies on the relationships of the variables studied with the training needs for effective and efficient work performance in the discussion than in the introduction (where almost none are referenced). It would be convenient to transfer them to the introduction to justify their inclusion in the study, as well as the bibliography on which their relevance is based. I believe that, if these changes are made, the work will gain in scientific quality, as corresponds to those published in a journal like this one.

Author Response

This study analyzes the correlations between socio-emotional competence (SEC) and academic performance (AA) among nursing students and compares the students' level of each SEC core skill (critical thinking disposition, self-directed learning, creativity, emotional intelligence, problem solving and collaboration) and academic performance (clinical performance and subjective academic performance), with the ultimate goal of reflecting on a necessary educational program at SEC for nursing students. It is a work as interesting as it is necessary, however, it could be improved. Specifically, I propuse the following recommendations for publication.

Regarding the introduction, given that the central axis of the work is to justify the need to acquire socio-emotional competence (SEC) for nursing students, I consider it necessary to explain how the authors conceptualize the socio-emotional competence construct (SEC) on which they plan the study. We verify throughout the work that they incorporate different measures as components of the same, but they do not clarify why these and not others, nor do they indicate any theoretical model that encompasses them as part of the same construct. On the other hand, scientific evidence indicated in this section on which the study is justified are not referenced, for example, the paragraph that begins on line 68. They are developed in the discussion.

→ Thank you for your insightful comments on presenting the socio-emotional competence (SEC) model. The theoretical model of the socio-emotional competence (SEC) applied in this study was conceptualized and explained. In addition, previous studies that reported that socio-emotional competence (SEC) were related to academic achievement were described to explain the validity of the conceptual component of this study (page 2-3). And as reviewer’s comment, the cited references were described (page 3).

Regarding the description of the evaluation instruments, and in coherence with what I have just indicated, it is not clear why different instruments are used to measure a single construct. On what scientific and / or theoretical basis do you consider that the different variables evaluated correspond to the SEC construct. It would be necessary to explain the model on which they are based to indicate that the SEC is made up of the dimensions that they indicate as defining it.

→ We have been trying to provide a conceptual framework as recommended. But it was not feasible because the purpose of this study was to identify the canonical correlation between SEC and academic achievement. Thus, we made an every effort to clarify the purpose of this paper and to clarify the SEC construct throughput the paper instead of providing a conceptual framework.

On the other hand, in all the instruments described, it does not define what it actually measures, not how it obtains the measure it uses. In almost all of its describes that they are made up of different subscales, of which sometimes it reports Cronbach's α and other times it does not, or vice versa (as it does with the collaboration variable). You would have to indicate how you get the only punctuation you use, its meaning and its alpha.

→ We presented Cronbach's alpha for sub-scales in Critical thinking Disposition, Self-directed Learning, Emotional Intelligence, and Problem-solving.

Regarding the results section, given the imbalance in the number of components of some categories, and given the possible influence of these variables on the variables studied, it would be necessary to include them in the correlation analyzes and perform analysis of variance to check if there are statistically differences significant between the qualitative variables (for example gender and religion) and the dependent variables, reporting in both cases the effect size and specifying which in "small", "medium" and "large". statistically differences significant between the qualitative variables (for example gender and religion) and the dependent variables, reporting in both cases the effect size and specifying which in "small", "medium" and "large".

→ The contents of religious beliefs were deleted because they were not related to the purpose. I would like to politely ask what does the dependent variable you mentioned mean? In our study, the main concern was canonical correlation, and I heard an answer from a statistician as follows: “it was not a problem that the number of components of some categories of the two variables were different because it made a linear combination of each variable.” And also it was not the purpose of this study to identify the difference according to the general characteristics, the test was not performed. In this study, the effect size was the canonical correlation coefficient, 0.76. We added the statement indicating that the effect size was large (page 7).

Likewise, regarding the Discussion section, in coherence with what was commented in the introduction, it is striking that it is where more articles are incorporated that link the variables studied in the population studied. These articles should be transferred to the introduction to justify the reason for the proposed research, and the reason for the incorporation in the study of the variables studied in the type of students studied (nursing). That is, we find more studies on the relationships of the variables studied with the training needs for effective and efficient work performance in the discussion than in the introduction (where almost none are referenced). It would be convenient to transfer them to the introduction to justify their inclusion in the study, as well as the bibliography on which their relevance is based.

→ In the introduction, the SEC conceptual model of this study was explained, and previous studies were additionally described. In addition, the discussion was supplemented by citing additional previous studies (page 2-3, page 8-9).

 I believe that, if these changes are made, the work will gain in scientific quality, as corresponds to those published in a journal like this one.

Thank for your reviewing.

Round 2

Reviewer 1 Report

Thank you for accepting the reviewers' comments and trying to revise the manuscript according to them. I will accept it in the updated version.

Reviewer 3 Report

-